# VLKEB: A Large Vision-Language Model Knowledge Editing Benchmark

**Han Huang**[1,2*]   **Haitian Zhong**[2*]   **Tao Yu**[2]   **Qiang Liu**[2†]

**Shu Wu**[2]   **Liang Wang**[2]   **Tieniu Tan**[2,3]

[1]University of Chinese Academy of Sciences (UCAS)
[2]New Laboratory of Pattern Recognition (NLPR),
State Key Laboratory of Multimodal Artificial Intelligence Systems (MAIS),
Institute of Automation, Chinese Academy of Sciences (CASIA)
[3]Nanjing University

## Abstract

Recently, knowledge editing on large language models (LLMs) has received considerable attention. Compared to this, editing Large Vision-Language Models (LVLMs) faces extra challenges from diverse data modalities and complicated model components, and data for LVLMs editing are limited. The existing LVLM editing benchmark, which comprises three metrics (Reliability, Locality, and Generality), falls short in the quality of synthesized evaluation images and cannot assess whether models apply edited knowledge in relevant content. Therefore, we employ more reliable data collection methods to construct a new Large **V**ision-**L**anguage Model **K**nowledge **E**diting **B**enchmark, **VLKEB**, and extend the Portability metric for more comprehensive evaluation. Leveraging a multi-modal knowledge graph, our image data are bound with knowledge entities. This can be further used to extract entity-related knowledge, which constitutes the base of editing data. We conduct experiments of different editing methods on five LVLMs, and thoroughly analyze how do they impact the models. The results reveal strengths and deficiencies of these methods and hopefully provide insights for future research. The codes and dataset are available at: https://github.com/VLKEB/VLKEB.

## 1 Introduction

With the rapid advancement and widespread deployment of LLMs, knowledge editing has emerged as an important topic [1–4]. The knowledge stored within LLMs can suffer from issues such as errors, deficiencies and obsolescence. Knowledge editing aims to efficiently correct and update this information while ensuring minimal impact on unrelated content. Numerous LLM editing methods have been proposed [5–14] and benchmarks have been established to assess these approaches. Metrics such as *Reliability*, *Generality*, *Locality* and *Portability* are commonly used [2] in these benchmarks. Reliability defines a reliable edit where the post-edit model produces the target answer for a given case. Generality requires the post-edit model to understand equivalent neighbors, such as rephrased sentences. Locality emphasizes localized editing, ensuring that the output of unrelated knowledge is retained. Portability evaluates whether models effectively apply edited knowledge to relevant content. These methods and benchmarks have greatly advanced the research in LLM knowledge editing.

In contrast, the task of knowledge editing for LVLMs has not been extensively studied. Currently, only one benchmark (MMEdit [15]) explores LVLM editing. This benchmark extends Reliability,

---

*Equal contribution.   ✉ han.huang@cripac.ia.ac.cn ✉ haitian.zhong@cripac.ia.ac.cn
†Corresponding author.   ✉ qiang.liu@nlpr.ia.ac.cn

38th Conference on Neural Information Processing Systems (NeurIPS 2024) Track on Datasets and Benchmarks.

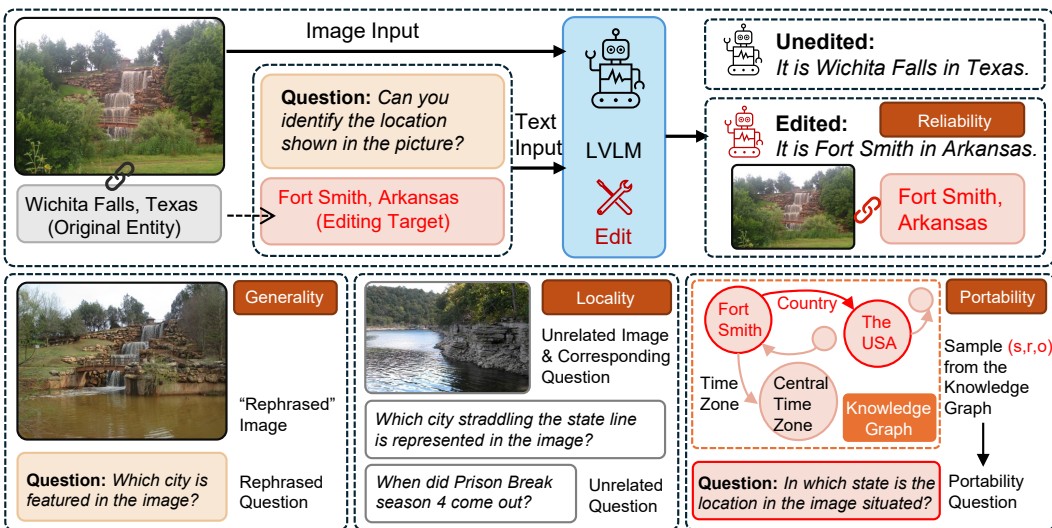

Figure 1: The image belongs to *"Wichita Falls"* originally and the editing target is to make LVLM recognize it as *"Fort Smith"*. The answer from LVLM measures the edit **Reliability**. The **Generality** inputs are "rephrased" images (*i.e.* belong to the same entity but different in perspective or appearance) and rephrased questions. **Locality** inputs are unrelated images and questions. **Portability** inputs are generated from sampled triples containing editing entity *'Fort Smith'* from the knowledge graph.

Generality and Locality, and adapts several editing methods from LLMs to LVLMs. However, it has some limitations. First, it uses synthesized images in Generality evaluation, which are generated by Stable Diffusion [16] from image captions. This could result in content inconsistency with original images and side-effect of less accurate evaluations. Second, the lack of evaluation of Portability is a significant gap, as effective use of edited knowledge is crucial to deploying an edited model in realistic applications. Furthermore, the limited quantity of data in this field poses a challenge to the progress of LVLM editing; therefore, additional data can greatly benefit the development of this area.

To address these issues, we have developed a new *Large Vision-Language Model Knowledge Editing Benchmark*, named **VLKEB**, designed to assess and improve the capabilities of knowledge editing methods in the field of LVLMs. An LVLM editing case using VLKEB is illustrated in Fig.1.

In our study, we source data from the multi-modal knowledge graph MMKG [17], which includes images linked to knowledge entities. The presence of multiple images for each entity in MMKG makes it practicable to selectively choose image pairs for assessing the Generality. During the selection process, we choose clear and representative image pairs, ensuring that each pair contains a same entity but presented in varied perspectives or appearances. For the Reliability test, corresponding entities of chosen images are employed. Subsequently, we filter similar images from the remaining entities (none of their images were chosen previously) to construct the image Locality test. We utilize GPT to generate questions and answers during data construction.

Another important advancement of our work is the extension of the Portability metric for more comprehensive evaluation. As shown in Fig.1, we sample relational triples $(s, r, o)$ of editing-involved entities from knowledge graphs to construct test examples. The edited entities serve as the subjects of these sampled triples, which are then used to generate portability questions.

Compared with the MMEdit benchmark, our benchmark offers several advantages and exhibits key differences. Firstly, our image selection prioritizes real images, mitigating potential flaws present in synthesized images. Secondly, we extend Portability evaluation which reveals the ability of methods to make edited model effectively use edited knowledge in related contents. Lastly, by using a multi-modal knowledge graph as our source distinctly associates each image with a specific entity, enhancing the clarity of what knowledge it carries, and is expansible by incorporating other knowledge in diverse relation triples as test cases. The key differences are presented in Tab.1.

In summary, our main contributions are as follows:

Table 1: Differences between **MMEdit** and **VLKEB**. (Rel: Reliability; T-*: Text-*; I-*: Image-*.)

| Comparison | | MMEdit | VLKEB (Ours) |
|---|---|---|---|
| **Metrics** | Rel/T-Gen/T-Loc
I-Generality
I-Locality
Portability | Yes
AI-Generated, less controllable
Random sample, easier evaluation
No Evaluation | Yes
Real images, manual check
Filtered sample, harder evaluation
Yes |
| **Construct** | Data Source | VQAv2 & COCO Caption; No entity connections, No Portability | MMKG; extensive connections, enhanced Portability evaluation |
| | Image Quality | Real & synthetic images; factual flaws observed in synthetic images | Real images in the datasets; quality guarantee, factually accurate |

- We introduce a new benchmark VLKEB, which is specifically designed for evaluating LVLM knowledge editing. The quality of data is guaranteed and it mitigates the challenge of limited data in this research area.

- Our work extends the pivotal metric of Portability into the field of LVLM knowledge editing, providing a more comprehensive assessment of the models' ability to transfer and apply edited knowledge effectively.

- We conducted experiments on various LVLMs using different editing methods. These experiments contribute valuable insights into the performance and limitations of existing knowledge editing approaches on LVLMs.

## 2 Related Works

**LLM Editing Benchmarks** The widely used datasets for LLM editing are ZsRE [18] and COUN-TERFACT [5]. ZsRE utilizes reading-comprehension examples for relation-slot filling tasks sourced primarily from the WikiReading dataset [19]. The COUNTERFACT dataset evaluates the ability of robustly learn new facts by challenging models with complex factual associations rather than simple lexical changes. The MQuAKE [20] dataset further evaluates knowledge editing generalization by focusing on multi-hop questions, which challenge models to navigate through interconnected information accurately. RippleEdits [21] complements this by testing the consistency of knowledge updates across related facts, highlighting the complexity of maintaining coherent knowledge.

**LLM Editing Methods** As the baseline, fine-tuning adjusts specific layers of language models or vision modules. Cutting-edge methods offer efficient solutions for knowledge updates. For example, MEND [6] leverages low-rank decomposition of gradients that enables rapid and targeted updates to knowledge and minimizes degradation on other inputs. SERAC [7] introduces a novel approach by integrating an explicit memory system. Using a scope classifier to determine the relevance of cached edits, SERAC ensures dynamic updates while maintaining the integrity of the base model. IKE (In-Context Knowledge Editing) [8] proposes an unsupervised retriever that constructs demonstrations to inject new factual knowledge without direct parameter updates. By ranking demonstrations based on their similarity to the editing target, IKE offers a scalable and efficient way to update information. Some of the LLM editing methods are adapted to LVLM in this study.

**Large Vision-Language Models** LVLMs involve aligning modalities during pre-training and refining response generation through instruction-based tuning, significantly improving their ability to handle complex multi-modal tasks. For instance, mPLUG-Owl [22] enhances multi-modal capabilities through a two-stage training approach and low-rank adaption [23], achieving superior performance. LLaVA [24] emphasizes pre-training and fine-tuning an alignment network alongside Vicuna, while Qwen-VL [25] introduces innovative features like a visual receptor and a three-stage training pipeline, excelling in visual-centric tasks and dialogue communication. We conduct experiments on various LVLMs to assess the performance of different editing methods.

# 3 Dataset Construction

**Problem Formulation**    An LVLM editing dataset $\mathcal{D}_{\text{edit}} = \{(i_{\text{e}}, x_{\text{e}}, y_{\text{e}}, y'_{\text{e}})_i\}$ contains image input $i_{\text{e}}$, text input $x_{\text{e}}$, ground truth $y_{\text{e}}$ and editing target $y'_{\text{e}}$. Given $i_e$ and $x_e$, an unedited LVLM produces $f(i_e, x_e; \theta) = y_e$, where parameters $\theta = \theta_{\text{vision}} \times \theta_{\text{text}}$. After knowledge editing, the edited LVLM with $\theta'$ are expected to successfully change the original outputs $f(i_e, x_e; \theta') = y'_e$.

## 3.1   Metrics: Reliability, Generality, Locality and Portability

To evaluate the knowledge editing methods, key evaluation criteria from prior work [15] are Reliability, Generality and Locality. Besides, based on our benchmark, we introduced Portability metric.

**Reliability** measures the proportion of target answers that the edited model can produce correctly.

**Generality** assesses how well the model responds to neighboring concepts in two modalities.

**Locality** measures how much of the stored knowledge, unrelated to the edit cases, remains unchanged in the edited model by comparing the outputs of the unedited and edited models.

**Portability**   Knowledge is interconnected, so modifying one fact affects related facts, complicating editing evaluation. Since model knowledge is not isolated, adjustments must consider broader implications. Portability evaluates if edited knowledge can be effectively applied to related content:

$$\mathcal{M}_{\text{portability}} = \mathop{\mathbb{E}}_{\substack{(i_e, x_e, y_e, y'_e) \sim \mathcal{D}_{\text{edit}} \\ (x_p, y_p) \sim \mathcal{P}(i_e, x_e, y_e, y'_e)}} \mathbb{1}\left\{ f(i_e, x_p; \theta') = y_p \right\},$$

where $x_p$ and $y_p$ are text inputs and outputs that belong to $\mathcal{P}(i_e, x_e, y_e, y'_e)$, which denotes the portability scope given the input $i_e, x_e, y_e$, and target output $y'_e$.

## 3.2   Construction Process

**Preparation**   The raw data in MMKG are knowledge triples $(s, r, o)$, where s is the subject, r is the relation, and o is the object, with image URLs for each entity. We start from an image $i$ and its corresponding entity $e$. The chosen LVLM is edited to associate $i$ with another entity $e'$. The editing process forms an editing triple $(i, e \rightarrow e')$. In the example ([a picture of Messi], Messi→Ronaldo), it aims to modify the knowledge within an LVLM to make it interpret the person in the picture as Ronaldo. We manually construct a prompt template $t_i(e; \mathcal{R})$ for an image $i$ and its entity $e$ and use GPT (see appendix) to generate a questions $i$ with given relation set $\mathcal{R}$ (could be empty), to which the answer is precisely $e$.

**Image Selection**   We start from image selection, recognizing the significance of image quality in evaluation. We retrieve the available images with URLs in MMKG and remove duplicates. We then utilize the pre-trained CLIP model [26] as an image feature extractor to assess similarities within each set of images corresponding to entities. Pairs of images with high similarity scores are then manually inspected to confirm that they belong to the same entity but differ in perspective or appearance.

**Reliability, Generality and Locality Evaluation Data Construction**   We choose Visual-Question-Answering (VQA) as the evaluation task. We start by using maximum weight bipartite matching to identify pairs of entities $(e, e')$ with the highest similarity based on their shared relations. These chosen pairs form a set of editing triples $(i, e \rightarrow e')$ as previously mentioned.

Next, we manually review generated QA to make sure the questions and answers match. For each entity and image, we generate two questions $q_1$ and $q_2$, which can be viewed as rephrase of each other. We put one in *Reliability* set $\mathcal{D}_{\text{edit}} = \{(i, q_1, e, e')_i\}$ and another in *Text Generality* set. For *Image Generality* test, the previous image pair selection enables us to get "rephrased" image of edited one. For *Text Locality*, we follow MMEdit to choose NQ dataset [27] which contains unrelated QAs and randomly put them in $\mathcal{D}_{\text{loc}}^{text}$. For *Image Locality*, we filter similar images for edit-involved images from unedited entities. We then generate questions and answers for each filtered image and entity. The test data in $\mathcal{D}_{\text{loc}}^{image}$ are thus similar in both modalities to that in $\mathcal{D}_{\text{edit}}$, making it more difficult for LVLM editing methods to perform well than random image locality test.

**Portability Evaluation Data Construction**  As mentioned, Portability represents whether editing methods can apply edited knowledge within the portability scope $\mathcal{P}\left(i_e, x_e, y_e, y_e'\right)$. For example, after performing the edit ([a picture of Messi], Messi→Ronaldo), we expect the LVLM to understand that the individual in the image, now identified as C. Ronaldo, is playing in Saudi Arabia instead of the USA. This means the model can apply the edited knowledge to related questions.

During the construction of the portability data, we select the connected triples of an edited entity, forming one-hop reasoning portability queries. For example, given $(i, e \rightarrow e')$ and $(s_1, r_1, o_1)$, where $e' = s_1$, the connected triple could be formulated as $\langle(i, e \rightarrow e'), (s_1, r_1, o_1)\rangle$. We then collect and generate questions $t_i(o_1; r_1)$ to form one-hop Portability evaluation dataset.

Nevertheless, Under more complex scenarios, it remains uncertain if an edited model can effectively process multi-hop reasoning tasks related to the edit. We suggest evaluating edited models with multi-hop reasoning VQA task by considering a connected chain of knowledge triples $\mathcal{C} = \langle(i, e \rightarrow e'), (s_1, r_1, o_1), \ldots, (s_n, r_n, o_n)\rangle$, where $e' = s_1$ and the object of the $i$-hop knowledge should serve as the subject of the subsequent knowledge in the sequence, i.e., $o_i = s_{i+1}$. We then similarly generate questions $t_i(o_n; r_1, \ldots, r_n)$ and incorporate them into the portability scope.

**Dataset Summary**  In total, the dataset contains 8174 editing cases with 18434 images. We split them into train and test set of 5000 and 3174 cases respectively. See appendix for more details.

# 4 Experiments

We conduct experiments on five LVLMs, which are BLIP2 [28], MiniGPT-4 [29], mPLUG-Owl2 [30], Qwen-VL [25] and LLaVA-1.5 [24]. The detailed versions of these LVLMs are in appendix. We test *single editing* and *sequential editing*[31, 11, 32]. *Single editing* updates a single knowledge once at a time and then evaluates editing results. This is a common setting of knowledge editing. In contrast, *sequential editing* continuously updates knowledge. Their difference is illustrated in Fig.2.

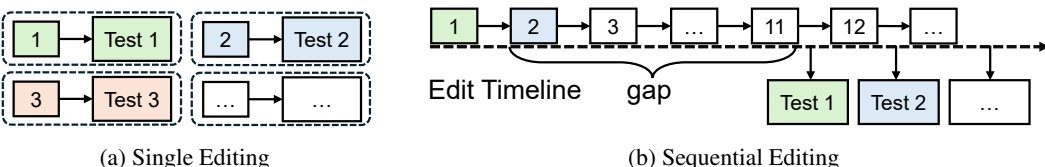

(a) Single Editing                                    (b) Sequential Editing

Figure 2: In Fig.2a, the single editing takes one edit at a time and evaluate immediately, while in Fig.2b the sequential editing involves continuous edits and test after several other edits.

## 4.1 Editing Methods and Experiment Settings

**Fine-tune (FT)**  We choose different parts of models during FT: the LLM layers or vision module.

**Knowledge Editor (KE)** [9] trains a bidirectional-LSTM as hyper network, which predicts weight updates of specified model parameters with gradients and condition input $\{(y_e \rightarrow y_e')|x_e\}$. We choose last layers of LLMs as the parameters to be updated in editing process.

**IKE** [8] does not change model parameters. It retrieves and builds similar demonstrations from training set, and inject new knowledge by prompting. This process is consistent across all models.

**SERAC** [7] is a memory-based method that has a scope classifier model and a counterfactual model. In our experiments, the classifier is trained from a BERT model [33] and the counterfactual model differs across LVLMs, which is set to be the LLM used by corresponding LVLM.

**MEND** [6] method enables models to efficiently update the parameters of the last layers in LLMs within LVLMs, utilizing low-rank gradient decomposition coupled with predictive parameter updates.

## 4.2 Single Editing Results and Findings

**High Reliability and Generality: In single editing, memory-based methods benefit from having only one piece of new knowledge stored, while parameter-update methods fit the single new**

Table 2: The single editing results of various editing methods applied to different LVLMs.
Rel.: Reliability; T/I-Gen.: Text/Image Generality; T/I-Loc.: Text/Image Locality; Port.: Portability

| Model | Method | Rel.↑ | T-Gen.↑ | I-Gen.↑ | T-Loc.↑ | I-Loc.↑ | Port.↑ |
|---|---|---|---|---|---|---|---|
| **BLIP2-OPT** ($\sim$ 3.8 B) | FT (LLM) | **99.75** | 99.08 | 98.95 | 71.10 | 19.90 | 17.13 |
| | FT (Vis) | 99.33 | 96.68 | 99.13 | 99.99 | 5.30 | 27.22 |
| | KE | 94.45 | 92.40 | 93.34 | 64.13 | 12.22 | 34.73 |
| | IKE | 99.47 | **99.40** | **99.56** | 70.11 | 10.26 | **44.22** |
| | SERAC | 96.02 | 95.99 | 96.01 | **100.0** | 2.40 | 15.25 |
| | MEND | 98.52 | 98.42 | 98.47 | 99.34 | **89.05** | 28.80 |
| **MiniGPT-4** ($\sim$ 7.8 B) | FT (LLM) | 99.60 | 98.72 | 98.10 | 90.17 | 35.39 | 27.13 |
| | FT (Vis) | **100.0** | 84.89 | 99.19 | **99.99** | 20.26 | 37.06 |
| | KE | 98.47 | 97.89 | 98.11 | 75.47 | 16.14 | 48.06 |
| | IKE | 99.98 | **99.68** | **99.98** | 59.25 | 9.73 | **54.30** |
| | SERAC | 99.37 | 97.30 | 99.29 | 99.93 | 4.54 | 49.22 |
| | MEND | 99.20 | 98.98 | 99.15 | 99.46 | **92.67** | 40.09 |
| **LLaVA-1.5** ($\sim$ 7 B) | FT (LLM) | 99.59 | 99.43 | 99.31 | 86.34 | 29.24 | 30.23 |
| | FT (Vis) | 99.80 | 99.12 | 97.55 | **99.99** | 18.79 | 54.43 |
| | KE | 99.07 | 97.59 | 98.65 | 77.36 | 15.25 | 48.62 |
| | IKE | **99.99** | 99.66 | **100.0** | 68.65 | 14.25 | **63.33** |
| | SERAC | 99.93 | **99.78** | 99.93 | 99.98 | 1.91 | 45.03 |
| | MEND | 99.54 | 99.21 | 99.52 | 99.36 | **90.14** | 40.39 |
| **Qwen-VL** ($\sim$ 9.7 B) | FT (LLM) | 97.92 | 96.30 | 95.48 | 72.80 | 37.23 | 16.15 |
| | FT (Vis) | **100.0** | 95.27 | 62.28 | **100.0** | 14.14 | 30.61 |
| | KE | 98.71 | 98.70 | 98.26 | 72.09 | 52.63 | 42.10 |
| | IKE | 99.01 | 98.85 | **99.01** | 57.97 | 10.48 | **57.99** |
| | SERAC | 97.62 | 95.68 | 97.84 | 99.85 | 0.81 | 38.22 |
| | MEND | 99.54 | **99.36** | 97.76 | 97.75 | **78.65** | 32.35 |
| **mPLUG-Owl2** ($\sim$ 8.2 B) | FT (LLM) | 99.21 | 95.72 | 99.39 | 71.42 | 34.31 | 42.77 |
| | FT (Vis) | 97.24 | 96.36 | 82.39 | **99.99** | 50.14 | **74.09** |
| | KE | 89.10 | 88.37 | 88.62 | 55.80 | 21.07 | 46.82 |
| | IKE | **99.98** | **99.93** | **100.0** | 64.88 | 16.59 | 64.83 |
| | SERAC | 99.03 | 97.73 | 98.99 | 99.97 | 1.32 | 48.52 |
| | MEND | 98.65 | 98.15 | 94.26 | 99.56 | **90.47** | 37.68 |

**piece of knowledge well.** In Tab.2, most methods and models achieve nearly 100% accuracy in Reliability due to the single editing test setting. As each edit is separately tested and the test is right after each edit, memory-based methods like SERAC retrieve the single new knowledge (with the "highest" input similarity) and append it to the test inputs. Similarly, IKE appends the edit case as a "New Fact" prompt before each test. Parameter-update methods like FT, KE, and MEND fit the new knowledge at each edit, leading to high Reliability. Additionally, LLMs exhibit high Text Generality due to their ability to handle rephrased questions, and the well pre-trained vision encoder ensures high Image Generality by effectively handling similar images.

**Varied Locality: Memory is a double-edged sword and parameter updates also harm models.** SERAC exhibits divergent Locality in text and image in Tab.2, with nearly $100\%$ in T-Loc but the highest I-Loc is merely $4.5\%$. The causes of this phenomenon are twofold. First, Text Locality data is collected from another irrelevant dataset (Sec.3.2) which is totally different from the edit data; therefore, it is classified to have very low similarity and processed by the original model. Second, Image Locality involves similar texts to edit cases, causing SERAC to misclassify and forward them to the counterfactual model, producing different outputs against the original model. IKE always builds in-context examples and appends edit text as a new fact before the test, which greatly affects model outputs. IKE's lower I-Loc compared to T-Loc is due to appended edits misdirecting the models with

false facts. In FT, we observe that fine-tuning different parts yields distinct results. Fine-tuning vision modules (FT-Vis) has higher T-Loc than fine-tuning LLM layers (FT-LLM), indicating that FT-LLM has greater and more direct impact on model outputs as it modifies the final LLM layers. In contrast, FT-Vis has lower I-Loc in because it changes the vision encoder or projector parameters, affecting the visual ability to distinguish between similar images. Comparing MEND with KE, MEND has overall better locality performance. This could be attributed to two reasons: First, KE predicts parameter updates based on gradients and text condition input, while MEND relies directly on token-wise activations and gradients, providing richer information about which parameters are crucial for an update [6]. Second, MEND includes a locality constraint during training which KE lacks, helping to maintain Locality. MEND shows the best average locality but it is imperfect as the outputs of edited models still differ somewhat from unedited ones.

**Portability: Can the model effectively apply the edited knowledge?** From the Portability column in Tab.2, we find that IKE generally achieves higher results than other methods, except in the case of mPLUG-Owl2 where FT-Vis has highest results. This demonstrates how prompting with examples and providing the necessary new knowledge can help models answer portability questions. In single editing, SERAC always appends edited knowledge before test, effectively acting as if IKE has only one demonstration. Consequently, SERAC shows inferior Portability results compared to IKE. In contrast, FT, KE and MEND which change parameters related to edits, do not account for the interconnected knowledge, resulting in generally poor results. The unsatisfactory outcomes suggest that these editing methods can not effectively utilize edited knowledge, as SERAC and IKE rigidly require edit texts as prompts, and FT, KE and MEND perform case-centric updates.

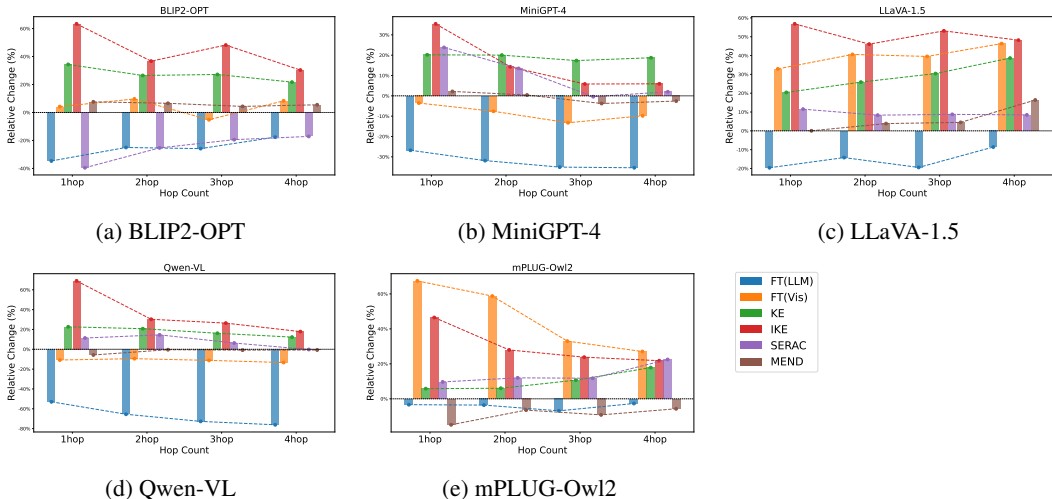

| (a) BLIP2-OPT | (b) MiniGPT-4 | (c) LLaVA-1.5 |
| (d) Qwen-VL | (e) mPLUG-Owl2 | |

Figure 3: Relative change (compared with unedited base model) of Multi-hop Portability results.

### 4.3 Multi-hop Portability: Performance Degradation Across Hops

Our experiments in Sec.4.2 evaluate the performance of edited models on the one-hop Portability dataset. As described in Sec.3.2, the Portability dataset includes elements of multi-hop reasoning VQA tasks. Consequently, we also conduct experiments on the multi-hop Portability evaluation dataset to determine whether the edited models can utilize knowledge in more complex scenarios. As illustrated in Fig.3, we displayed the multi-hop Portability results using the relative change compared to base (unedited) model, *i.e.* Relative Change (%) $= (\frac{\text{Portability}-\text{Base Portability}}{\text{Base Portability}})$.

**Portability Metrics Decline with Increasing Number of Hops** In our experiments, we observe a consistent decline in portability metrics as the number of hops increases. This decline is observed across nearly all models and editing methods. This phenomenon can be attributed to the escalating complexity of reasoning required to accurately apply the edited knowledge across multiple interconnected facts. Each additional hop introduces new intermediate entities and relationships, compounding the difficulty for the model to maintain correct and consistent reasoning.

**IKE performs well across all models, while Fine-tuning generally performs poorly, especially Fine-tuning LLM heads.** Since IKE consistently builds in-context examples and appends edit text as a new fact before the test, it maintains a robust understanding of the edited knowledge across different scenarios. This method's strength lies in its ability to dynamically incorporate new information in a contextually relevant manner, which enhances its performance even in multi-hop reasoning tasks. The in-context learning approach allows IKE to adapt to the complexities introduced by additional hops, thereby preserving the accuracy and consistency of the edited knowledge.

In contrast, Fine-tuning LLM heads tends to perform poorly across all models. The process of fine-tuning involves directly modifying the model parameters, which can lead to overfitting on the edited examples and a failure to generalize to related but unedited contexts. As the number of hops increases, the fine-tuned models struggle to apply the edited knowledge accurately due to the rigid and localized nature of the parameter updates. Interestingly, while the fine-tuning of visual modules (FT-Vis) also performs suboptimally, it does not suffer as severely as the fine-tuning of LLM heads. The closer proximity of the LLM heads to the output layer increases the likelihood of overfitting, as the updates directly impact the final predictions.

### 4.4 Sequential Editing: Performance Degradation Due to Forgetting and Confusion

Editing knowledge separately is impractical in real-world scenarios, as knowledge changes over time, necessitating continuous updates to our models. We test sequential editing (Fig.2b) on FT-LLM, FT-Vis, SERAC and MEND, while exclude KE and IKE because they require the edit data as part of the input during test time, which is not feasible in practical applications. The results are in Fig.4.

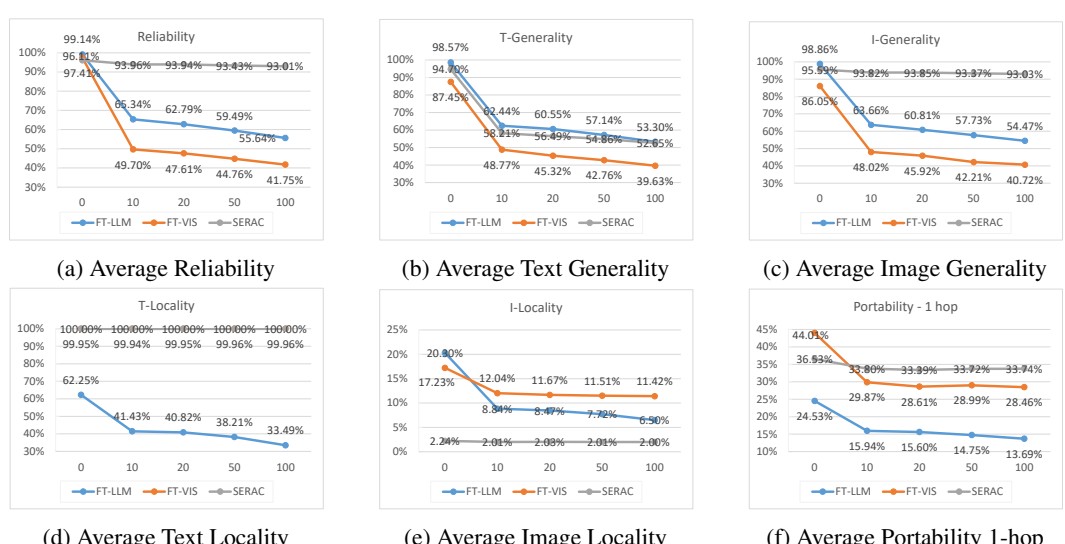

(a) Average Reliability      (b) Average Text Generality      (c) Average Image Generality

(d) Average Text Locality      (e) Average Image Locality      (f) Average Portability 1-hop

Figure 4: Average results in sequential editing. Horizontal axis is the test gap number in Fig.2b.

MEND is not included in these figures because, after a certain number of edits, we observed "NaN" values in the logits of model outputs, indicating a collapse. For example, Blip2-OPT outputs "NaN" after 50 edits, MiniGPT-4 after around 14 edits and LLaVA after around 20 edits. MEND learns to predict parameter updates based on gradients of edits and parameters of original model during training. However, in sequential editing, the corresponding LLM parameters change continuously and differ from the unedited model in the training phase, making MEND unable to predict precise updates, ultimately leading to collapse. Similar findings were observed by Han et al. [32].

Apart from MEND, other methods exhibit different characteristics compared to single editing. **First, FT tends to forget previous edits as model parameters change.** The downward trends in Reliability and Generality for FT in Fig.4a4b4c indicate that preserving previous edits becomes increasingly difficult with a larger test gap. On the contrary, **SERAC does not forget due to explicit memory cache but becomes confused when memory is filled with more edits**. SERAC maintains consistency in Reliability and Image Generality because exact test texts are stored from previous edits. However,

Text Generality involves rephrased text that is not stored, challenging the similarity retrieval within SERAC and leading to incorrect results due to false retrievals.

Fig.4d shows that FT-Vis tends to preserve model outputs in unrelated test text, as it leaves the LLM layer unchanged, while FT-LLM does the opposite. In Fig.4e, both FT-Vis and FT-LLM exhibit low scores since the test texts are similar to edits, with FT-LLM declining further as the gap increases. These Locality results suggest that **parameter updates can harm the model, and updating the LLM has greater impact than updating the vision module**.

In Fig.4f, Portability also decreases with an increasing gap. As shown in Sec.4.3, the unedited model can predict some correct logits according to input texts through a forward pass. The average Portability of base models is $36.80\%$, and the FT Portability drops lower after gap $= 10$, indicating that **edited models cannot effectively apply edited knowledge to related content**. Therefore, to investigate if Portability can be improved, we experiment with 1-hop portability in Sec.4.5.

### 4.5 Edit One-Hop Knowledge: Room for Portability Improvement

Table 3: Portability increases after additionally edit corresponding one-hop knowledge.

| Portability | FT (LLM) | FT (Vis) | SERAC | MEND |
|---|---|---|---|---|
| **Blip2-OPT** | 17.13→46.71 (↑ 29.58) | 27.22→39.00 (↑ 11.78) | 16.16→32.12 (↑ 15.96) | 28.75→68.18 (↑ 39.43) |
| **MiniGPT-4** | 27.13→44.65 (↑ 17.52) | 37.06→56.91 (↑ 19.85) | 47.49→51.10 (↑ 3.61) | 39.19→53.83 (↑ 14.64) |
| **LLaVA-1.5** | 30.23→74.79 (↑ 44.56) | 54.43→84.81 (↑ 30.38) | 45.03→72.75 (↑ 27.72) | 40.39→70.73 (↑ 30.34) |
| **Qwen-VL** | 16.15→88.88 (↑ 72.73) | 30.61→55.15 (↑ 24.54) | 38.22→54.44 (↑ 16.22) | 32.35→69.41 (↑ 37.06) |
| **mPLUG-Owl2** | 42.77→59.37 (↑ 16.60) | 74.09→93.80 (↑ 19.71) | 48.52→67.91 (↑ 19.39) | 37.68→70.71 (↑ 33.03) |

In this tentative exploration, we consider editing the second triple (Sec.3.2) of Portability data (*i.e.* first edit $(i, e \rightarrow e')$ and then edit $(e', r, o)$).We generate QAs for the triples and the model twice before conducting the Portability test, and the results are in Tab.3.The table displays Portability before and after additionally editing 1-hop knowledge, indicated by the arrows. All Portability results improved, with some showing huge gains. We investigate the improvement of SERAC by comparing retrieved results. We find that after editing 1-hop knowledge, some Portability test texts have higher similarity to 1-hop knowledge QA stored in memory. In these cases, the 1-hop knowledge is appended before the test text, leading to higher Portability. In other words, **SERAC does not actually apply the first edit** $(i, e \rightarrow e')$ **to the model but only the second triple** $(e', r, o)$ **that contains Portability answers**. This unintended behavior means that the models are not aware of the first edit and are indeed given a "cheat sheet" in Portability test. For FT and MEND, parameter updates fit the 1-hop knowledge and help the model provide more correct answers. **This may suggest that Portability can be improved by explicitly incorporating it into the training phase**. However, the challenges faced by these parameter-update methods in sequential editing still need careful consideration.

## 5 Conclusion, Limitation and Future Direction

We establish a knowledge editing benchmark for LVLMs and evaluate diverse editing methods across various models. Our analysis delves into the impact of these methods on the models, revealing both strengths and weaknesses. These findings offer valuable insights for potential future directions.

**Direct LVLM Editing** In this work, we evaluate LLM editing methods, but the the search for an efficient LVLM editing method is ongoing. These methods are adapted from LLM techniques and are not specifically designed for LVLMs, as they do not account for the interaction between modalities. research could explore direct LVLM editing methods to address this gap.

**Sequential Editing in LVLM** We have observed performance degradation across methods in sequential editing. Since this test closely mirrors real-world scenarios, future work on LVLM editing should focus on mitigating these issues.

**Portability Evaluation** Current methods do not adequately consider Portability, resulting in unsatisfactory evaluation results. Our preliminary experiments show that there is room for Portability improvement. Future research should further emphasize Portability as an important aspect.

## Acknowledgments and Disclosure of Funding

This work is jointly sponsored by National Natural Science Foundation of China (62236010, 62141608, 62206291).

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

## A  Dataset Details

Dataset and Croissant metadata are available at https://www.kaggle.com/datasets/hymanh/vlkeb-data. The images are sorted by entity IDs in subfolders. Additionally, four JSON files of text inputs are provided, along with descriptions on the site. The license is also included.

The statistics for VLKEB are presented in Tab.4. VLKEB includes a total of 8174 edits, divided into 5000 for training and 3174 for evaluation. There are 18434 images used in the Reliability, Generality, and Locality tests. The Portability test utilizes the same images as the Reliability test and comprises a total of 4819 cases. These cases are distributed among 1-hop, 2-hop, 3-hop, and 4-hop categories, with 1278, 1238, 1193, and 1110 cases, respectively. The comparison with MMEdit is in Tab.5.

Table 4: Statistics of VLKEB.

|        | All (train/eval)  |          | Rel.  | Gen.  | Loc.  |
|--------|-------------------|----------|-------|-------|-------|
| **#Edits** | 8174 (5000/3174) | **#Images** | 8172 | 6627 | 3635 |
|        | **All (eval only)** | **1-hop** | **2-hop** | **3-hop** | **4-hop** |
| **#Port.** | 4819 | 1278 | 1238 | 1193 | 1110 |

Table 5: Comparison of statistics between MMEdit and VLKEB (ours).

|                          | MMEdit | VLKEB |
|--------------------------|--------|-------|
| **#Edits (VQA)**         | 8439   | 8174  |
| **#Edits (Image Caption)** | 3849 | -     |
| **#Portability**         | -      | 4819  |
| **#Real Images**         | 11857  | 18436 |
| **#Synthesized Images**  | 12288  | -     |

VLKEB contains a total of 8796 entities (in edits and Image Locality). Since each entity can belong to multiple classes, we provide the proportion of each class in Fig.5. These entities belong to different categories, contributing to the diversity of the data.

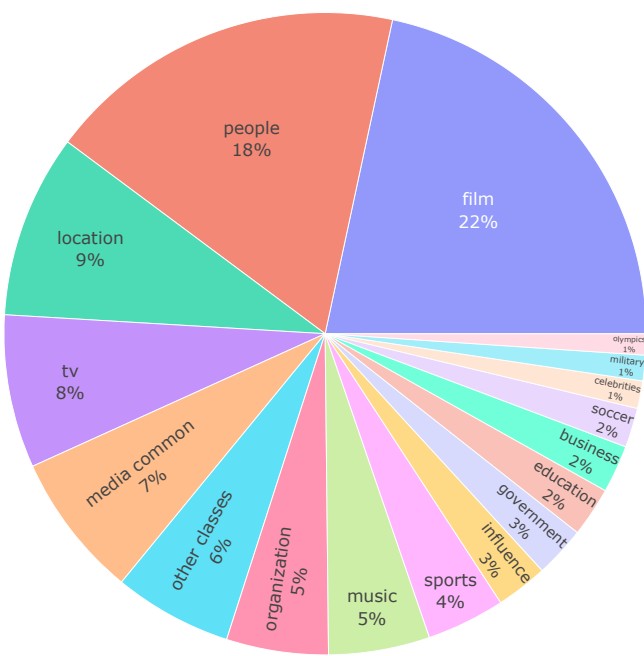

Figure 5: The class proportions of entities.

# B Experiment Details

## B.1 Metrics

To evaluate the knowledge editing methods, we used key evaluation criteria from prior work [15]: Reliability, Generality and Locality. Also, we introduced Portability metric.

In the following formulas, $f$ is the LVLM model that maps the input (text and image) to the output parameterized by $\theta$, while $\theta'$ refers to the edited model parameters.

**Reliability**

$$\mathcal{M}_{\text{reliability}} = \mathbb{E}_{(i_e, x_e, y_e, y'_e) \sim \mathcal{D}_{\text{edit}}} \mathbb{1}\left\{ f\left(i_e, x_e; \theta'\right) = y_e \right\},$$

where $\mathcal{D}_{\text{edit}}$ represents the original editing dataset.

**Generality**

$$\mathcal{M}_{\text{generality}}^{\text{text}} = \mathbb{E}_{\substack{\left(i_e, x_e, y_e, y'_e\right) \sim \mathcal{D}_{\text{edit}} \\ x_r \sim \mathcal{N}(x_e)}} \mathbb{1}\left\{ f\left(i_e, x_r; \theta'\right) = y_e \right\},$$

$$\mathcal{M}_{\text{generality}}^{\text{image}} = \mathbb{E}_{\substack{(i_e, x_e, y_e) \sim \mathcal{D}_{\text{edit}} \\ i_r \sim \mathcal{N}(i_e)}} \mathbb{1}\left\{ f\left(i_r, x_e; \theta'\right) = y_e \right\},$$

where the $\mathcal{N}(x_e)$ and $\mathcal{N}(i_e)$ stand for the rephrased neighborhood of input text and image respectively.

**Locality**

$$\mathcal{M}_{\text{locality}}^{\text{text}} = \mathbb{E}_{(x_l, y_l) \sim \mathcal{D}_{\text{loc}}^{\text{text}}} \mathbb{1}\left\{ f\left(x_l; \theta'\right) = f(x_1; \theta) \right\},$$

$$\mathcal{M}_{\text{locality}}^{\text{image}} = \mathbb{E}_{(i_l, x_l, y_l) \sim \mathcal{D}_{\text{loc}}^{\text{image}}} \mathbb{1}\left\{ f\left(i_l, x_l; \theta'\right) = f(i_l, x_l; \theta) \right\},$$

where $\mathcal{D}_{\text{loc}}^{\text{text}}$ and $\mathcal{D}_{\text{loc}}^{\text{image}}$ are locality datasets.

**Portability**

$$\mathcal{M}_{\text{portability}} = \mathbb{E}_{\substack{\left(i_e, x_e, y_e, y'_e\right) \sim \mathcal{D}_{\text{edit}} \\ (x_p, y_p) \sim \mathcal{P}\left(i_e, x_e, y_e, y'_e\right)}} \mathbb{1}\left\{ f\left(i_e, x_p; \theta'\right) = y_p \right\},$$

where $\mathcal{P}\left(i_e, x_e, y_e, y'_e\right)$ denotes the Portability scope given input $i_e, x_e, y_e$ and target output $y'_e$.

## B.2 The versions of LVLMs

We conduct experiments on five LVLMs, and their specific version are in Tab.6.

Table 6: LVLM versions in experiments. (Vis.: Vision Encoder)

| LVLM | BLIP2-OPT | MiniGPT-4 | LLaVA-1.5 | mPLUG-Owl2 | Qwen-VL |
|------|-----------|-----------|-----------|------------|---------|
| LLM | OPT-2.7B | Vicuna-7B | Vicuna-7B-v1.5 | LLaMA-2-7B | Qwen-7B |
| Vis. | ViT-g (1B) | ViT-g (1B) | ViT-L (0.3B) | ViT-L (0.3B) | ViT-G (1.9B) |

**BLIP2-OPT** BLIP2-OPT is a variant of the BLIP2 framework that leverages the power of large language models with 2.7 billion parameters (opt-2.7b). It is a pre-trained model specifically designed for image description and related tasks. The model is available at: https://github.com/salesforce/LAVIS/tree/main/projects/blip2.

**MiniGPT-4** MiniGPT-4 is an open-source chatbot with image understanding capabilities. It is built upon the foundation of the LLM like Vicuna and the BLIP-2 LVLM. By aligning a frozen visual encoder with a frozen LLM, Minigpt4 achieves efficient multi-modal capabilities. The model is available at: https://github.com/Vision-CAIR/MiniGPT-4.

**LLaVA-1.5**  LLaVA-1.5 is a multi-modal pre-trained model that demonstrates exceptional capabilities in cross-modal understanding and generation. It marks a significant advancement over previous versions, enabling it to handle a wider range of tasks with higher complexity. The model is available at: https://github.com/haotian-liu/LLaVA.

**Qwen-VL**  Qwen-VL is a visual-language model developed by Alibaba. It aims to provide advanced capabilities in visual understanding, localization, text reading, and beyond. The model is available at: https://github.com/QwenLM/Qwen-VL.

**mPLUG-Owl2**  mPLUG-Owl2 is a multi-modal large language model developed by Alibaba. This advanced model leverages modality collaboration to significantly enhance the performance of both text-based and multi-modal tasks. Its unique modular design and modality adaptation mechanism make it stand out among its peers. The model is available at: https://github.com/X-PLUG/mPLUG-Owl/tree/main/mPLUG-Owl2.

### B.3  Knowledge Editing Methods

**FT (Fine-tune)**  updates model parameters by performing gradient descent on the chosen model parameters. We save the target layers' current weights for later restoration at single editing. An AdamW optimizer is configured to ensure that only those target fine-tuning parameters' gradients are computed and updated.

**MEND (Model Editor Networks with Decomposition) [6]**  enables efficient update of the parameters in LLMs within LVLMs. It operates by training a set of small auxiliary networks, or model editor networks, that use a single desired input-output pair to make targeted, local adjustments to a model's behavior without affecting its overall performance on unrelated tasks. MEND leverages the low-rank structure of fine-tuning gradients, allowing it to parameterize the gradient transformation in a computationally efficient manner, even for models with billions of parameters.

**SERAC (Semi-Parametric Editing with a Retrieval-Augmented Counterfactual) [7]**  is a memory-based method that has a scope classifier and a counterfactual model. The scope classifier is trained to classify if an input falls into the editing scope. If so, the cached memory of related edit is put together into the counterfactual model; otherwise, the input is sent to the original model. In our experiments, the classifier is trained from a BERT model [33] and the counterfactual model differs across LVLMs, which is set to be the LLM used by corresponding LVLM.

**KE (Knowledge Editor) [9]**  trains a bidirectional-LSTM as hyper network, which predicts weight updates of specified model parameters with gradients and condition input $\{(y_e \rightarrow y'_e) | x_e\}$. The hyper-network takes the input of the original model's parameters, along with the fact to be modified, and predicts the required updates to these parameters. We choose last layers of LLMs in editing.

**IKE (In-Context Knowledge Editing) [8]**  does not change model parameters. It retrieves and builds similar demonstrations from training set, and inject new knowledge by prompting. This process is consistent across all models. The text in training set is formatted to "New Fact: {question} {answer}\nPrompt: {question} {answer}\n\n" and preprocessed into embeddings.

### B.4  Experiment Resources and Parameters

In this study, we utilize an internal cluster equipped with NVIDIA A100 80GB GPUs, and we employ PyTorch in our experiments. The parameters are in the following and in config files in code repository.

**FT-LLM**

| Models | Steps | Edit Layer | Optimizer | Edit LR |
|---|---|---|---|---|
| BLIP2-OPT | 15 | $31^{st}$ layer of Transformer Module | AdamW | $2e-4$ |
| MiniGPT-4 | 15 | $31^{st}$ layer of Transformer Module | AdamW | $1e-4$ |
| LLaVA-1.5 | 10 | $31^{st}$ layer of Transformer Module | AdamW | $1e-4$ |
| Qwen-VL | 20 | $31^{st}$ layer of Transformer Module | AdamW | $1e-4$ |
| mPLUG-Owl2 | 20 | $31^{st}$ layer of Transformer Module | AdamW | $1e-4$ |

**FT-Vis**

| Models | Steps | Edit Layer | Optimizer | Edit LR |
|---|---|---|---|---|
| BLIP2-OPT | 15 | Qformer | AdamW | $2e-4$ |
| MiniGPT-4 | 15 | Qformer | AdamW | $1e-4$ |
| LLaVA-1.5 | 10 | mm_projector | AdamW | $1e-4$ |
| Qwen-VL | 25 | $47^{th}$ layer of ViT Module | AdamW | $2e-3$ |
| mPLUG-Owl2 | 25 | Visual Encoder | AdamW | $1e-3$ |

**MEND**

| Models | MaxIter | Edit Layer | Optimizer | LR |
|---|---|---|---|---|
| BLIP2-OPT | 40000 | layer $29, 30, 31$ of Transformer Module | Adam | $1e-6$ |
| MiniGPT-4 | 40000 | layer $29, 30, 31$ of Transformer Module | Adam | $1e-6$ |
| LLaVA-1.5 | 40000 | layer $29, 30, 31$ of Transformer Module | Adam | $1e-6$ |
| Qwen-VL | 40000 | layer $29, 30, 31$ of Transformer Module | Adam | $1e-6$ |
| mPLUG-Owl2 | 45000 | layer $29, 30, 31$ of Transformer Module | Adam | $1e-6$ |

**SERAC**

| Models | MaxIter | Edit Layer | Optimizer | LR |
|---|---|---|---|---|
| BLIP2-OPT | 50000 | all layers of OPT-125M | Adam | $1e-5$ |
| MiniGPT-4 | 50000 | $31^{st}$ layer of Vicuna-7B | Adam | $5e-5$ |
| LLaVA-1.5 | 50000 | $31^{st}$ layer of Vicuna-7B-v1.5 | Adam | $1e-5$ |
| Qwen-VL | 20000 | $31^{st}$ layer of Qwen-7B | Adam | $5e-5$ |
| mPLUG-Owl2 | 20000 | $31^{st}$ layer of LLaMA-2-7B | Adam | $5e-5$ |

**KE**

| Models | MaxIter | Edit Layer | Optimizer | LR |
|---|---|---|---|---|
| BLIP2-OPT | 30000 | layer $29, 30, 31$ of Transformer Module | RMSprop | $3e-4$ |
| MiniGPT-4 | 30000 | layer $29, 30, 31$ of Transformer Module | RMSprop | $3e-4$ |
| LLaVA-1.5 | 30000 | layer $29, 30, 31$ of Transformer Module | RMSprop | $3e-4$ |
| Qwen-VL | 30000 | $31^{st}$ layer of Transformer Module | RMSprop | $3e-4$ |
| mPLUG-Owl2 | 30000 | $31^{st}$ layer of Transformer Module | RMSprop | $3e-4$ |

**IKE**  Use the sentence-transformers model all-MiniLM-L6-v2 to embed texts and retrieve similar edits in training set. The number of demonstrations is set to 32 for all models.

## C  More Details of Dataset Construction

### C.1  Construction Process

**Image Selection**  In the image selection process (Sec.3.2), we write a simple tool to show image pairs with high similarity, as shown in Fig.6. Then we manually check if they belong to a same entity and have different views. The result is recorded each time one of the buttons is clicked.

**Identify Similar Entities**  In Sec.3.2, we explain that we identify pairs of entities with the highest similarity. The purpose of this process is to ensure $e$ and $e'$ belong to the same class so that an edit $(i, e \rightarrow e')$ makes sense. We extract the relation set for each entity from the FB15K knowledge graph and count the number of shared relations between every pair of entities. These counts become the weights of the edges in a bipartite graph, where both sets of nodes represent all entities. We then use a maximum weight bipartite matching algorithm to determine the optimal way to match these entities.

**Portability Construction**  The knowledge triples for Portability test are extracted from DB15K knowledge graph and the relations of these extracted triples are presented in Tab.7.

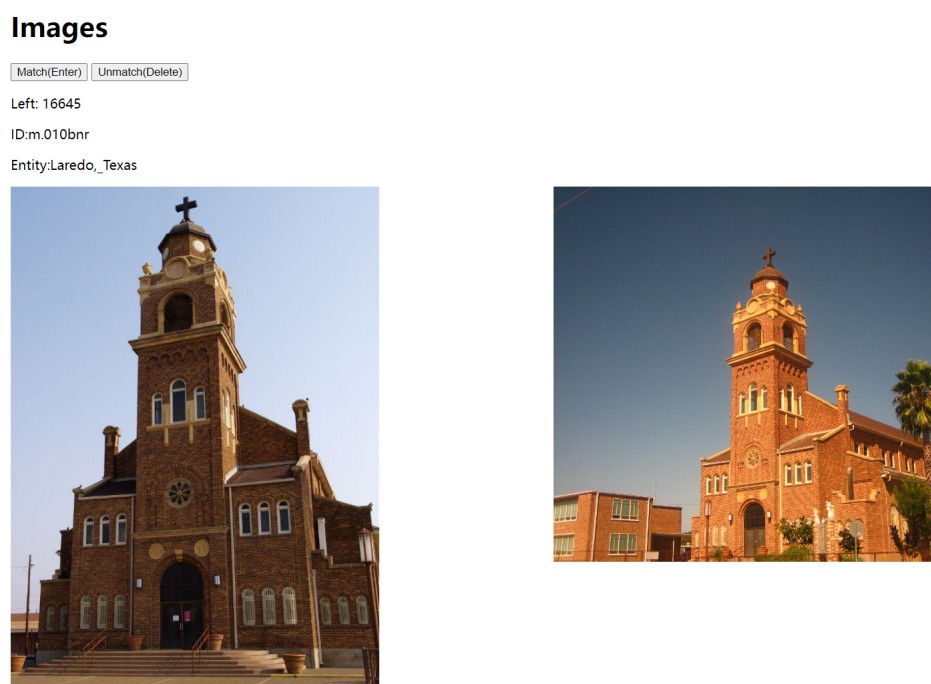

Figure 6: A simple tool to select similar image pairs for Image Generality test.

Table 7: Relations in Portability data construction.

| | | | | |
|---|---|---|---|---|
| affiliation | countySeat | headquarter | musicSubgenre | presenter |
| almaMater | creator | hometown | narrator | producer |
| associatedBand | currency | instrument | nationality | region |
| birthPlace | deathCause | language | network | religion |
| campus | deathPlace | largestCity | occupation | residence |
| capital | distributingCompany | leaderName | officialLanguage | spouse |
| channel | distributor | leaderParty | owner | state |
| child | editing | location | owningCompany | timeZone |
| citizenship | education | locationCity | parentCompany | writer |
| city | executiveProducer | locationCountry | party | |
| company | foundationPlace | musicComposer | place | |
| country | genre | musicFusionGenre | position | |

## C.2 Prompt Template for QA Generation

The prompt template for QA generation is in Tab.8 and Tab.9. We use gpt-3.5-turbo and gpt-4-turbo for the QA generation respectively.

## C.3 Examples of Each Test

Here we provide examples for single editing, multi-hop portability and edit one-hop knowledge.

Fig.7 shows three examples of single editing. The Text Generality input differs from Reliability input in question, while Image Generality differs in image. Text Locality input has text only, while Image Locality input has both image and text. Tab.10 shows an example of multi-hop Portability. Tab.11 shows an example of edit one-hop knowledge.

Table 8: The template of using ChatGPT (gpt-3.5-turbo) to generate Q&As for entities.

**System:**
You are a powerful question and answer generator.  Users have an image
and corresponding entity.  Given only the entity name, you should ask two
proper questions of what entity (person, location, sign, scene, poster,
logo, sight, occupation and so on) is shown in image, and the answer should
be exactly the entity name.  Output Format:  'Q: question1\nQ: question2\nA:
answer'.
**Example User:** Entity:  Denton,_Texas
**Example Assistant:**  Q: What city of Texas is depicted in the image?\nQ: Which
Texas city is seen in the picture?\nA: Denton

*[4 in-context demonstrations abbreviated]*

**User:**
Entity:  University_of_New_Mexico

**System:**

Q1:  What university is featured in the image?
Q2:  Which educational institution is depicted in the picture?
A: University of New Mexico.

Table 9: An example template of using ChatGPT (gpt-4-turbo-2024-04-09) to generate three-hop
Q&As from MMKG triples. We manually write 2 demonstrations as the prompt.

**System:**
You are a powerful three-hop question generator.  Users will provide a
chain of Multimodal Knowledge Graph (MMKG) triples, and you will help write
questions to ask about the entity in the picture based on the relation
chains.  You should choose one entity from the list if multiple entities
are present at the end.  The questions should focus on the head entity and
should not mention any of the entities in the triples explicitly.  Write
down the question and answer in the format:  'Q: question\nA: answer.'

*[2 in-context demonstrations abbreviated]*

**User:**
Given MMKG triples (Christina Aguilera; Katy Perry) (Katy Perry; spouse;
Russell Brand) (Russell Brand; birthPlace; United Kingdom) (United Kingdom;
language; English language).  Choose one entity from the list if multiple
entities at the end:  English language.  Ask questions about Christina
Aguilera in the picture without mentioning Christina Aguilera or Katy Perry
or Russell Brand or United Kingdom or English language.  Write down one
question and answer in natural English.  Format:  'Q: question\nA: answer.'

**System:**

Q: What is the country associated with the birthplace of the spouse of the
person shown in the picture?
A: English language

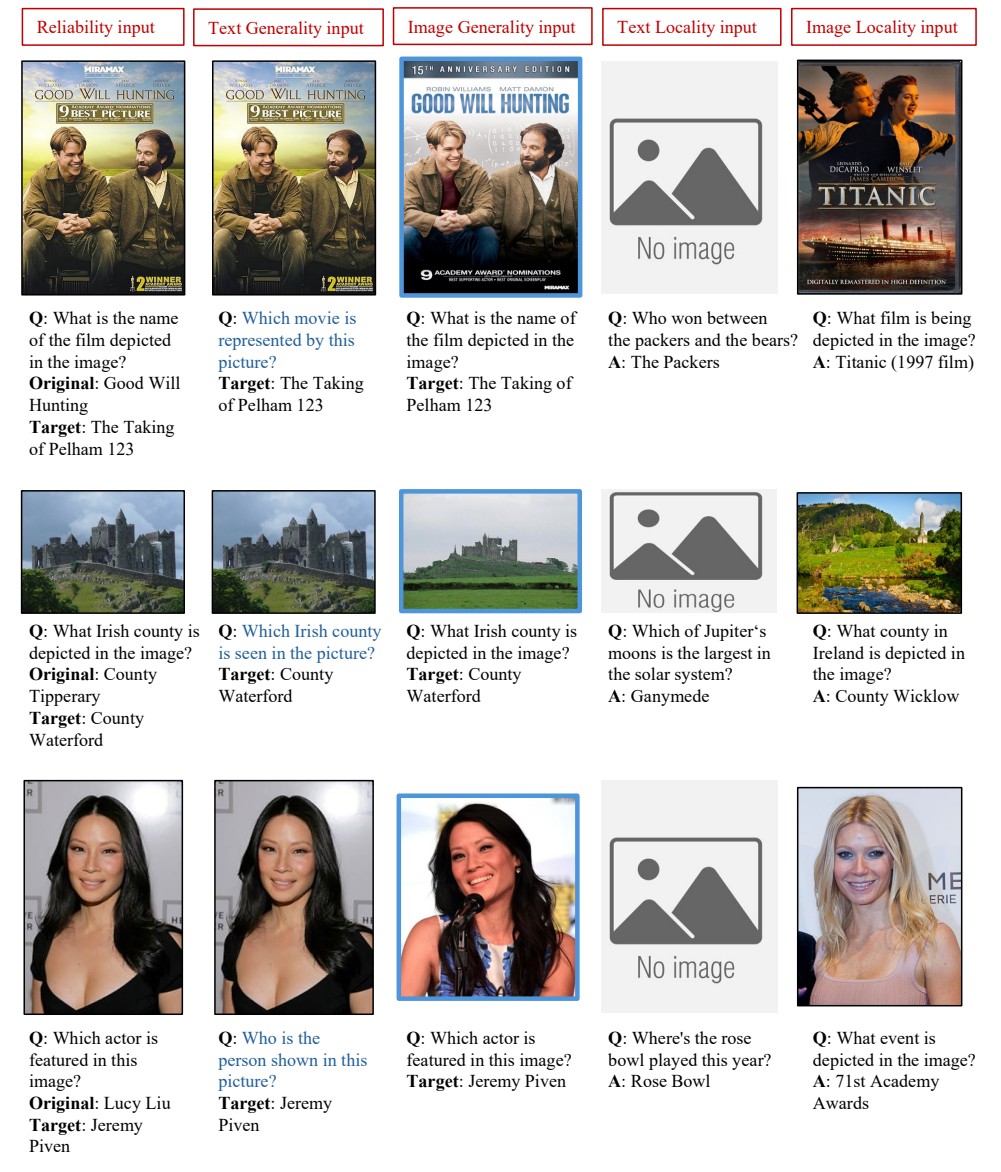

Figure 7: Three examples of single editing.

Table 10: An example of multi-hop Portability.

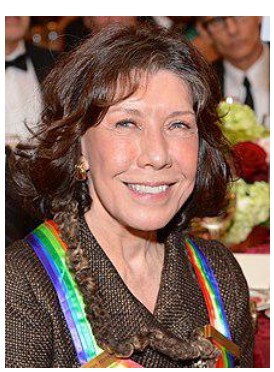

**Edit**
```
Q: Who is the actor featured in this image?
Target:  Andre Braugher
```

**1-hop triple:** (Andre Braugher, birthPlace, Chicago)
```
Q: What city is the birth place of the person in the picture?
A: Chicago
```

**2-hop triples:** (Andre Braugher, almaMater, Juilliard School), (Juilliard School, location, New York City)
```
Q: Where is the alma mater of the person associated with the person in the
picture located?
A: New York City
```

**3-hop triples:** (Andre Braugher, almaMater, Juilliard School), (Juilliard School, location, New York City), (New York City, timeZone, Eastern Time Zone)
```
Q: What is the time zone of the city where the alma mater of the person
connected to the individual in the picture is located?
A: Eastern Time Zone
```

**4-hop triples:** (Andre Braugher, residence, Chicago), (Chicago, country, United States), (United States, largestCity, New York City), (New York City, timeZone, Eastern Time Zone)
```
Q: What is the time zone of the largest city in the country where the
residence of the person associated with the entity shown in the picture
is located?
A: Eastern Time Zone
```

Table 11: An example of edit one-hop knowledge.

*<Same Image as Tab.10>*
**Edit**
```
Q: Who is the actor featured in this image?  Target:  Andre Braugher
```

**Edit 1-hop triple:** (Andre Braugher, birthPlace, Chicago)
**Edit** `Q: In which city was Andre Braugher born?`
```
Target:  Chicago
```

**Test 1-hop Portability:**
**Test** `Q: What city is the birth place of the person in the picture?`
```
A: Chicago
```

# D Original Results of Multi-hop Portability and Sequential Editing

In Sec.4.3, we present the results of multi-hop Portability in figures. Here we provide the original results in Tab.12. In Sec.4.4, we present the results of sequential editing in figures. Here we provide the original results in Tab.13.

Table 12: Original data of multi-hop Portability results.

| LVLM | method | 1-hop | 2-hop | 3-hop | 4-hop |
|---|---|---|---|---|---|
| **BLIP2-OPT** | base | 26.72 | 27.25 | 26.20 | 29.49 |
| | FT (LLM) | 17.46 | 20.43 | 19.43 | 24.28 |
| | FT (Vis) | 27.84 | 29.89 | 24.84 | 31.95 |
| | KE | 35.90 | 34.46 | 33.32 | 35.89 |
| | IKE | 43.65 | 37.25 | 38.83 | 38.43 |
| | SERAC | 16.16 | 20.35 | 21.11 | 24.44 |
| | MEND | 28.75 | 29.04 | 27.34 | 31.11 |
| **MiniGPT-4** | base | 38.36 | 38.26 | 40.88 | 42.12 |
| | FT (LLM) | 28.09 | 26.07 | 26.55 | 27.22 |
| | FT (Vis) | 37.00 | 35.35 | 35.47 | 37.98 |
| | KE | 46.10 | 45.92 | 47.97 | 50.01 |
| | IKE | 51.94 | 43.73 | 43.26 | 44.61 |
| | SERAC | 47.49 | 43.42 | 40.71 | 42.96 |
| | MEND | 39.19 | 38.42 | 39.34 | 41.03 |
| **LLaVA-1.5** | base | 40.38 | 38.06 | 36.58 | 36.25 |
| | FT (LLM) | 32.46 | 32.65 | 29.47 | 33.13 |
| | FT (Vis) | 53.67 | 53.51 | 51.04 | 53.05 |
| | KE | 48.62 | 47.91 | 47.71 | 50.26 |
| | IKE | 63.33 | 55.59 | 56.01 | 53.71 |
| | SERAC | 45.03 | 41.23 | 39.78 | 39.32 |
| | MEND | 40.39 | 39.53 | 38.22 | 42.19 |
| **Qwen-VL** | base | 34.32 | 35.06 | 36.29 | 42.50 |
| | FT (LLM) | 16.15 | 12.09 | 9.90 | 10.14 |
| | FT (Vis) | 30.61 | 31.74 | 32.23 | 36.82 |
| | KE | 42.10 | 42.34 | 42.16 | 47.72 |
| | IKE | 57.99 | 55.15 | 53.29 | 56.23 |
| | SERAC | 38.22 | 40.16 | 38.58 | 42.41 |
| | MEND | 32.35 | 34.91 | 35.98 | 42.19 |
| **mPLUG-Owl2** | base | 44.25 | 40.18 | 38.68 | 38.27 |
| | FT (LLM) | 42.77 | 41.63 | 39.75 | 40.16 |
| | FT (Vis) | 74.09 | 68.55 | 56.76 | 52.44 |
| | KE | 46.82 | 45.79 | 47.23 | 48.64 |
| | IKE | 64.83 | 55.25 | 52.84 | 50.27 |
| | SERAC | 48.52 | 48.36 | 47.70 | 50.56 |
| | MEND | 37.68 | 40.39 | 38.76 | 38.92 |

Table 13: Original data of equential editing results.

| LVLM | method | gap | Rel. | T-Gen. | I-Gen. | T-Loc. | I-Loc. | Port. |
|---|---|---|---|---|---|---|---|---|
| **BLIP2-OPT** | **FT (LLM)** | - | 99.95 | 99.23 | 100.00 | 72.20 | 20.18 | 17.50 |
| | | 10 | 57.52 | 56.12 | 57.18 | 34.74 | 4.40 | 15.04 |
| | | 20 | 54.84 | 53.64 | 54.19 | 32.56 | 3.89 | 14.16 |
| | | 50 | 51.25 | 50.44 | 50.51 | 27.27 | 2.54 | 13.47 |
| | | 100 | 47.18 | 46.61 | 46.34 | 18.49 | 1.17 | 10.04 |
| | **FT (Vis)** | - | 99.53 | 96.89 | 99.27 | 100.00 | 5.87 | 27.90 |
| | | 10 | 27.99 | 29.42 | 27.86 | 100.00 | 1.33 | 21.76 |
| | | 20 | 30.31 | 29.52 | 30.31 | 100.00 | 1.36 | 21.70 |
| | | 50 | 30.09 | 28.94 | 30.01 | 100.00 | 1.27 | 24.35 |
| | | 100 | 30.80 | 29.69 | 30.67 | 100.00 | 1.23 | 23.59 |
| | **SERAC** | - | 90.42 | 89.23 | 90.50 | 100.00 | 2.47 | 12.62 |
| | | 10 | 90.30 | 41.34 | 90.25 | 100.00 | 2.47 | 13.19 |
| | | 20 | 90.47 | 38.81 | 90.58 | 100.00 | 2.58 | 12.49 |
| | | 50 | 90.43 | 35.71 | 90.43 | 100.00 | 2.52 | 12.81 |
| | | 100 | 90.54 | 35.43 | 90.54 | 100.00 | 2.46 | 13.30 |
| | **MEND** | N/A | | | | | | |
| **MiniGPT-4** | **FT (LLM)** | - | 100.00 | 100.00 | 100.00 | 90.38 | 35.74 | 27.73 |
| | | 10 | 74.52 | 72.70 | 71.60 | 67.73 | 10.06 | 21.83 |
| | | 20 | 70.83 | 70.48 | 68.97 | 67.76 | 9.38 | 21.01 |
| | | 50 | 66.34 | 65.86 | 64.63 | 63.57 | 8.03 | 20.79 |
| | | 100 | 62.75 | 62.46 | 61.76 | 59.70 | 6.00 | 19.79 |
| | **FT (Vis)** | - | 99.77 | 87.46 | 99.77 | 100.00 | 21.62 | 42.16 |
| | | 10 | 45.47 | 47.04 | 45.57 | 100.00 | 16.01 | 37.53 |
| | | 20 | 44.55 | 47.17 | 44.50 | 100.00 | 15.96 | 36.28 |
| | | 50 | 41.02 | 45.21 | 41.77 | 100.00 | 15.79 | 37.27 |
| | | 100 | 40.12 | 43.16 | 40.33 | 100.00 | 15.75 | 36.53 |
| | **SERAC** | - | 96.24 | 95.30 | 96.18 | 99.90 | 4.46 | 40.50 |
| | | 10 | 96.12 | 58.38 | 96.05 | 99.90 | 4.49 | 40.38 |
| | | 20 | 96.02 | 56.77 | 95.95 | 99.90 | 4.47 | 39.44 |
| | | 50 | 96.02 | 55.58 | 95.95 | 99.93 | 4.44 | 39.31 |
| | | 100 | 95.52 | 54.72 | 95.42 | 99.93 | 4.45 | 40.20 |
| | **MEND** | N/A | | | | | | |
| **LLaVA-1.5** | **FT (LLM)** | - | 99.85 | 99.30 | 99.80 | 86.96 | 30.35 | 30.70 |
| | | 10 | 88.58 | 81.63 | 83.94 | 62.99 | 14.66 | 29.06 |
| | | 20 | 85.20 | 77.98 | 80.65 | 63.19 | 14.17 | 29.38 |
| | | 50 | 76.75 | 70.15 | 71.59 | 60.34 | 13.36 | 28.65 |
| | | 100 | 72.27 | 65.90 | 68.28 | 56.97 | 11.80 | 28.85 |
| | **FT (Vis)** | - | 100.00 | 98.98 | 98.75 | 100.00 | 19.87 | 55.48 |
| | | 10 | 95.55 | 93.61 | 88.33 | 100.00 | 3.08 | 47.98 |
| | | 20 | 91.46 | 88.53 | 83.66 | 100.00 | 2.89 | 46.75 |
| | | 50 | 83.08 | 80.78 | 71.70 | 100.00 | 2.64 | 47.00 |
| | | 100 | 69.58 | 67.28 | 66.68 | 100.00 | 2.53 | 45.93 |
| | **SERAC** | - | 98.56 | 97.46 | 98.56 | 99.96 | 1.94 | 41.46 |
| | | 10 | 98.56 | 74.64 | 98.56 | 99.96 | 1.93 | 40.63 |
| | | 20 | 98.56 | 72.97 | 98.56 | 99.96 | 1.92 | 40.43 |
| | | 50 | 98.56 | 72.36 | 98.56 | 99.96 | 1.92 | 40.80 |
| | | 100 | 98.56 | 70.32 | 98.56 | 99.96 | 1.91 | 41.08 |
| | **MEND** | N/A | | | | | | |

| Model | Method | | | | | | | |
|---|---|---|---|---|---|---|---|---|
| **Qwen-VL** | **FT (LLM)** | - | 96.26 | 96.64 | 95.21 | 70.33 | 35.46 | 15.95 |
| | | 10 | 61.50 | 57.98 | 60.60 | 11.53 | 1.68 | 7.11 |
| | | 20 | 57.96 | 55.99 | 56.49 | 10.34 | 1.49 | 5.56 |
| | | 50 | 54.07 | 52.17 | 52.97 | 8.95 | 1.18 | 5.61 |
| | | 100 | 49.06 | 46.59 | 48.57 | 6.92 | 0.80 | 5.18 |
| | **FT (Vis)** | - | 100.00 | 94.96 | 61.87 | 100.00 | 14.52 | 31.83 |
| | | 10 | 39.04 | 35.02 | 39.27 | 100.00 | 2.79 | 27.71 |
| | | 20 | 33.82 | 33.44 | 33.99 | 100.00 | 2.82 | 27.11 |
| | | 50 | 31.96 | 33.16 | 32.46 | 100.00 | 2.89 | 27.23 |
| | | 100 | 33.17 | 33.87 | 32.97 | 100.00 | 2.96 | 27.86 |
| | **SERAC** | - | 96.45 | 93.75 | 94.12 | 99.89 | 0.85 | 37.21 |
| | | 10 | 92.83 | 43.10 | 92.48 | 99.85 | 0.77 | 33.67 |
| | | 20 | 92.96 | 41.36 | 92.61 | 99.90 | 0.77 | 33.40 |
| | | 50 | 92.96 | 41.40 | 92.61 | 99.91 | 0.76 | 33.96 |
| | | 100 | 92.73 | 40.04 | 92.61 | 99.92 | 0.76 | 32.35 |
| | **MEND** | N/A | | | | | | |
| **mPLUG-Owl2** | **FT (LLM)** | - | 99.59 | 97.62 | 99.31 | 70.03 | 35.17 | 40.01 |
| | | 10 | 44.57 | 43.79 | 44.99 | 30.15 | 13.38 | 6.65 |
| | | 20 | 45.12 | 44.65 | 43.75 | 30.23 | 13.44 | 7.89 |
| | | 50 | 49.06 | 47.10 | 48.96 | 30.92 | 13.48 | 5.21 |
| | | 100 | 46.96 | 44.92 | 47.38 | 25.38 | 12.74 | 4.60 |
| | **FT (Vis)** | - | 99.97 | 97.42 | 82.31 | 99.99 | 50.37 | 74.06 |
| | | 10 | 40.43 | 38.78 | 39.06 | 100.00 | 36.97 | 14.39 |
| | | 20 | 37.89 | 27.93 | 37.16 | 100.00 | 35.34 | 11.22 |
| | | 50 | 37.63 | 25.71 | 35.12 | 100.00 | 34.96 | 9.10 |
| | | 100 | 35.09 | 24.15 | 32.97 | 100.00 | 34.61 | 8.41 |
| | **SERAC** | - | 98.89 | 97.76 | 98.61 | 99.98 | 1.46 | 50.87 |
| | | 10 | 91.98 | 73.58 | 91.78 | 100.00 | 0.40 | 41.11 |
| | | 20 | 91.68 | 72.53 | 91.54 | 100.00 | 0.39 | 41.19 |
| | | 50 | 89.19 | 69.23 | 89.30 | 100.00 | 0.40 | 41.70 |
| | | 100 | 87.68 | 62.76 | 88.01 | 100.00 | 0.41 | 41.76 |
| | **MEND** | N/A | | | | | | |

