# OpenReview forum: "VLKEB: A Large Vision-Language Model Knowledge Editing Benchmark"
_NeurIPS.cc/2024/Datasets_and_Benchmarks_Track — NeurIPS 2024 Track Datasets and Benchmarks Poster_

### Official Review · Reviewer_4DYX · 2024-07-18
**This paper presents VLKEB, a new large vision-language model knowledge editing benchmark. Compared to the existing MMEdit benchmark, this benchmark addressed the limitations in generality and introduced the evaluation of portability. Experimental results show the performance and influence of different editing methods on popular Vision-Language Models (VLMs).**

**Rating:** 6
**Confidence:** 4
**Correctness:** The details are overall sound and cor…
**Clarity:** This paper demonstrates clear writing…

**Review:**

Pros:
1. This paper is overall well-organised, with illustrative figures for ease of comprehension.
2. The comparison between the existing benchmark and this work is clearly presented, highlighting the contributions of this work.
3. The introduced benchmark can well benefit the field of VLM editing by complementing high-quality data and including comprehensive evaluation metrics.
4. The details of dataset construction, metrics and evaluations are clearly and formally introduced.
5. Comprehensive experiments are conducted to evaluate the performance of different editing methods in both single and sequential editing cases, along with a detailed analysis of the results.
Cons:
1. In Table 2, lots of methods achieve extremely high scores, nearly 100% accuracy, which seems to mean that this benchmark may not be challenging enough for modern methods.
2. Mathematical symbols should be introduced when they first appear for better clarity; for example, see Lines 109 – 110, Line 112, etc.
3. Figures 3 and 4 should be reorganised for better clarity and presentation.

**Strengths:**

1. This paper is overall well-organised, with illustrative figures for ease of comprehension.
2. The comparison between the existing benchmark and this work is clearly presented, highlighting the contributions of this work.
3. The introduced benchmark can well benefit the field of VLM editing by complementing high-quality data and including comprehensive evaluation metrics.
4. The details of dataset construction, metrics and evaluations are clearly and formally introduced.
5. Comprehensive experiments are conducted to evaluate the performance of different editing methods in both single and sequential editing cases, along with a detailed analysis of the results.

**Additional Feedback:**

N/A

**Documentation:**

This paper includes a detailed description of the dataset, metrics and evaluations. The source codes and the dataset can be accessed through the provided GitHub link.

**Limitations:**

Limitations and future directions are briefly discussed in Section 5.

**Opportunities For Improvement:**

Please revise and provide additional explanations regarding the cons mentioned in the review part.

**Relation To Prior Work:**

Yes.

**Summary And Contributions:**

This paper presents VLKEB, a new large vision-language model knowledge editing benchmark. Compared to the existing MMEdit benchmark, this benchmark addressed the limitations in generality and introduced the evaluation of portability. Experimental results show the performance and influence of different editing methods on popular Vision-Language Models (VLMs).

---

> ### Author Rebuttal · Authors · 2024-08-15
>
> **1.	High scores in Table 2:**
>
> Thank you for your review and for pointing out the high accuracy scores reported in Table 2. We appreciate your insights and would like to address your concerns as follows:
>
> **(i)** In the context of knowledge editing for LLMs and multimodal editing, **it is often observed that reliability and generality can be relatively high.** These metrics serve as foundational assurances rather than definitive indicators of a method's performance. Our observation is consistent with findings from relevant literature, which suggest that while reliability and generality are essential, they may not fully capture the complexities of knowledge editing tasks.
>
> **(ii) Introduction of More Challenging Metrics:** Recognizing the need for a more comprehensive evaluation, our approach specifically emphasizes additional, more challenging metrics. We have improved our data selection process for image locality and introduced portability as a more demanding metric. These enhancements are designed to test the methods in scenarios that more accurately reflect real-world applications, thus providing a deeper insight into the effectiveness of different editing strategies. This focus helps ensure that our benchmark not only tests basic competency but also probes the robustness and adaptability of editing methods.
>
> **2.	Writings and figures:**
>
> Thank you for your careful reading of our manuscript and for pointing out that the mathematical symbols and Figures 3 and 4 could be improved for better clarity. We agree that it is crucial for readers to have a clear understanding of the mathematical notations and figures.
>
> **(i)** We will add detailed definitions for the mathematical symbols as follows:
>
> Lines 109-110, $x_p$ and $y_p$ is the text input and outputs in portability test, and other symbols are consistent with those in Problem Formulation section.
>
> Line 112, in triple $(s, r, o)$, $s$ means subject, $r$ means relation and $o$ means object.
>
> We will check other symbols through the paper to make sure they are all clear.
>
> **(ii)** We will also update these figures by adjusting the layout to enhance clarity for better presentation.
>
> We appreciate your guidance on this matter, and we are committed to making these changes to enhance the readability of our paper.

---

### Official Review · Reviewer_agdD · 2024-07-24
**New benchmark on knowledge editing in large VLMs**

**Rating:** 6
**Confidence:** 3
**Correctness:** The claims seem to be correct.

**Review:**

**Pros**:
- The paper is relatively well-written and easy to follow. The research question is well-motivated, and the data generation process is well-documented;
- The experiments are relatively comprehensive by considering a diverse set of knowledge editing methods and settings;
- The analysis highlights several limitations of existing approaches with good summary;

**Cons**:
- Limited testing of knowledge editing that involves multimodal interactions (current work only seems to be a natural extension from the unimodal setting);
- Minor writing issues (see more below in Clarity).

**Strengths:**

Overall, the paper is generally well-written and easy to follow with well-motivated research question and clearly stated contributions. Knowledge editing remains an open challenge for LLM development and improvements of its evaluation is important. This paper proposed a new benchmark for evaluating knowledge editing in the multimodal domain, which currently has limited evaluation resources. The benchmark is well-documented with justified data generation procedures. The following analysis considers various knowledge editing methods and settings, exposing a wide range of limitations of existing methods.

**Additional Feedback:**

See comments above (Opportunities For Improvement).

**Clarity:**

Yes in general. The introduction is a bit too long (with repetitive definitions of the metrics and reiteration of the main contributions, which can be more concise) and bolded texts are a little bit over-used (terms can be italicized if needed). The numbers on the figures (e.g. Figure 3) can be larger.

**Documentation:**

Did not find any maintenance plan.

**Limitations:**

There is no limitation discussed in the last section Conclusion, Limitation and Future Direction.

**Opportunities For Improvement:**

The proposed extension of evaluation metric and new generation of more challenging data does not involve more testing of the effect of knowledge editing that involves multimodal interactions, for which the evaluation in multimodal setting is more complex than the unimodal setting. An example of such challenging multimodal interactions can be multiple back-and-forth references between image and text (multi-round QA). There can also be separate analysis for the effect of knowledge editing on both modalities (e.g. for locality, presenting only the rephrased question or the rephrased image).

**Relation To Prior Work:**

Yes, although in the related works section, the subsection Large Vision-Language Models do not seem necessary.

**Summary And Contributions:**

This paper proposed VLKEB as a new benchmark on knowledge editing in large vision-language models. The benchmark improves the data quality used to evaluate Reliability, Generality, Locality in prior work, and extends the Portability as a new metric. The following analysis shows various limitations of existing knowledge editing approaches for VLMs.

---

> ### Author Rebuttal · Authors · 2024-08-15
>
> **1.	Limited testing involving multimodal interactions like multi-round QA.**
>
> Thank you for your comments regarding the scope of our testing for knowledge editing, specifically concerning multimodal interactions. We appreciate the opportunity to clarify the focus and scope of our current research.
>
> **(i) Focus on Immediate Knowledge Editing:** Our research primarily concentrates on assessing the effectiveness of knowledge editing techniques. We evaluate whether the edited knowledge is correctly integrated within the models using targeted, single-round QA sessions. These sessions are specifically designed to measure key attributes such as reliability, generality, locality, and portability. While we recognize the complexity and potential of multimodal interactions in a multi-round setting, our study strategically prioritizes foundational aspects of knowledge editing. And multi-round QA often pertain more to historical memory and might be addressed through advancements in dialogue systems. Our focus ensures a robust evaluation of immediate edits without the confounding variables introduced by extended multi-round interactions.
>
> **(ii) Distinction Between Linguistic and Knowledge Capabilities:** Moreover, the capability for multi-round QA pertains more to the linguistic abilities of large models, whereas our research is centered on the knowledge contained within these models. These are distinct aspects. Our work specifically targets the latter, which does not inherently involve multi-round capabilities.
>
> In future research, it could be valuable to explore how models that have undergone knowledge editing can handle multi-round QA. However, this falls outside the current scope of our study, which is designed to assess the immediate effects of knowledge edits without the added complexity of interactive contexts.
>
> **2.	Separate analysis for the effect of knowledge editing on both modalities.**
>
> Thank you for suggesting an analysis of the separate effects of knowledge editing on each modality. We totally agree with this, and actually our experiments are already separated on both modalities. We assume it might be misleading in Figure 1 that rephrased questions and rephrased images are listed together. However, they are actually presented separately, which means that in image generality test, the inputs are rephrased images and original questions, and in text generality test, the inputs are original images and rephrased questions. We also provide more examples in Figure 7 in supplementary material, which may better illustrate how we run the experiments. We will further clarify this point in our paper.
>
> **3.	Limitations:**
>
> We recognize that the open knowledge editing methods and data used in our research could be exploited to manipulate the knowledge in LVLMs, potentially resulting in harmful outputs. Nevertheless, it's essential to note that our research employs publicly accessible datasets that comply with usage licenses, which helps address direct ethical concerns associated with the benchmark itself.
>
> **4.	Clarity:**
>
> Thank you for your constructive feedback on the clarity of our manuscript. We appreciate your suggestions and will implement the following revisions to address your concerns:
>
> We will streamline the introduction by consolidating the definitions of the metrics and summarizing the main contributions to eliminate redundancy. We will reduce the use of bolded text throughout the paper in line with your suggestion. We will increase the font size of the numbers in Figure 3 and other figures as necessary to improve readability.
>
> We hope these modifications will enhance the clarity and presentation of our paper. Thank you for guiding these improvements.

---

> > ### Comment · Reviewer_agdD · 2024-08-23
> >
> > Hi, thank you for the clarifications and revision to the paper. I think my original major concerns are mostly addressed and I will raise my rating from 5 to 6 and keep my confidence score unchanged.

---

### Official Review · Reviewer_wpwF · 2024-07-26
**A significant contribution to the knowledge editing benchmark for large vision-language models**

**Rating:** 8
**Confidence:** 3
**Correctness:** Yes
**Clarity:** Yes

**Review:**

This paper introduces the Vision-Language Knowledge Editing Benchmark (VLKEB), marking a significant advancement in the field of vision-language models. By establishing this new benchmark, the authors provide a standardized framework for evaluating and comparing various knowledge editing techniques, thereby facilitating a more systematic and consistent assessment process. The extension of the Portability metric further enhances the evaluation tool, allowing for a more comprehensive and in-depth analysis of performance across different models and methods.

A notable contribution of this work is the use of a multi-modal knowledge graph to bind image data with knowledge entities. This innovative approach enables the efficient extraction of entity-related knowledge, creating a robust foundation for editing data within vision-language models. By integrating visual and textual information in a cohesive manner, the authors have laid the groundwork for more sophisticated and accurate knowledge editing processes.

The authors conduct extensive experiments employing various editing methods on five different large vision-language models (LVLMs). Their thorough analysis of the impact of these methods on the models provides valuable insights into the strengths and weaknesses of current techniques. This detailed evaluation not only highlights the efficacy of different approaches but also identifies areas for future improvement, contributing significantly to the advancement of knowledge editing in vision-language models.

This paper is well-written and well-motivated, but the Figure 3 can be improved for better understanding.

**Strengths:**

This paper introduces the Vision-Language Knowledge Editing Benchmark (VLKEB), which represents a significant advancement in the field of vision-language models. By establishing a new benchmark, the authors provide a standardized framework for evaluating and comparing different knowledge editing techniques. Additionally, the extension of the Portability metric offers a more comprehensive evaluation tool, enhancing the depth and scope of performance assessments.

The use of a multi-modal knowledge graph to bind image data with knowledge entities is a noteworthy contribution. This approach enables the extraction of entity-related knowledge, forming a solid base for editing data.

The authors conduct extensive experiments using various editing methods on five different large vision-language models (LVLMs). Their in-depth analysis of how these methods impact the models provides valuable insights into the strengths and weaknesses of current techniques.

**Additional Feedback:**

NA

**Documentation:**

Yes

**Limitations:**

Yes

**Opportunities For Improvement:**

Figure 3 can be improved for better understanding.

Missing references:

Knowledge Editing for Large Language Models: A Survey

A Comprehensive Study of Knowledge Editing for Large Language Models

**Relation To Prior Work:**

Yes

**Summary And Contributions:**

This paper introduces a new benchmark for knowledge editing in large vision-language models, termed the Vision-Language Knowledge Editing Benchmark (VLKEB). Additionally, it extends the Portability metric for a more comprehensive evaluation. By leveraging a multi-modal knowledge graph, the image data are linked with knowledge entities, enabling the extraction of entity-related knowledge that forms the foundation of the editing data. The authors conduct experiments using various editing methods on five large vision-language models (LVLMs) and perform an in-depth analysis of their impact on the models. The results highlight the strengths and weaknesses of these methods, offering valuable insights for future research.

---

> ### Author Rebuttal · Authors · 2024-08-15
>
> Thank you for the positive evaluation and constructive feedback. We appreciate your suggestions for improving our manuscript.
>
> **1.	Improvement of Figure 3:**
>
> We acknowledge the feedback regarding Figure 3 and understand the need for clearer visual representation to enhance comprehension. We will revise this figure by enhancing the labeling and restructuring the layout to ensure that the information is communicated more effectively and clearly.
>
> **2.	Missing References:**
>
> We have reviewed these sources and agree that they are critical in this field. We will add these references to the relevant sections of our paper to provide a more comprehensive background and to acknowledge prior work in the field of knowledge editing for LLM.
>
> We hope these revisions will address your concerns and strengthen our submission. Thank you for bringing these issues to our attention.

---

> > ### Comment · Reviewer_wpwF · 2024-08-17
> >
> > Thank you for the author's response. I think this paper makes a clear contribution to the field, and I will maintain my original score.

---

### Official Review · Reviewer_Y4B2 · 2024-07-26
**This paper designs the VLKEB benchmark to evaluate the knowledge editing methods for Large Vision-Language Models (LVLMs). Its quality is marginally above acceptance threshold.**

**Rating:** 5
**Confidence:** 4
**Clarity:** Yes.

**Review:**

1. Quality: The paper is well-structured, and the experiments are thorough, providing a significant contribution to the field of LVLM knowledge editing.
2. Clarity: The paper is well written, with a logical flow of ideas and clear explanations of the methodology and results.
3. Originality: The introduction of the portability metric and the use of a multi-modal knowledge graph for data collection are original contributions that set this work apart.
4. Significance: The work is good, as it addresses a current gap in LVLM evaluation and provides a benchmark that can guide the future research.

**Strengths:**

1. Comprehensive Benchmark: The introduction of VLKEB fills the gap in the evaluation of LVLM knowledge editing by addressing limitations in existing benchmarks.
2. Portability Metric: Extending the portability metric provides a more holistic assessment of how well edited knowledge can be applied in relevant contexts.
3. Real Images: Using real images instead of synthesized ones improves the reliability of the evaluation.
4. Detailed Experiments: The experiments conducted on multiple LVLMs provide valuable insights into the performance of different knowledge editing methods.

**Additional Feedback:**

The paper is a good contribution to the field and could benefit from a discussion on the broader implications of the work. It will be helpful to include a section on future work and potential applications of the VLKEB benchmark.

**Correctness:**

The claims made in the submission are correct based on the information provided.

**Documentation:**

The paper includes sufficient details for reproducibility, with a clear explanation on data collection and organization.

**Ethics:**

The paper conforms to the ethics review guidelines, but it could benefit from a more explicit discussion of ethical concerns.

**Limitations:**

The paper does not extensively discuss the potential negative societal impacts of LVLM knowledge editing. The authors need to provide more details on the ethical considerations of the work.

**Opportunities For Improvement:**

1. Data Source Limitations: While MMKG is a robust data source, its coverage and diversity may still pose limitations for certain LVLM applications.
2. Manual Image Selection: The process of manually selecting and verifying images for the benchmark, although it is thorough, could be time-consuming and may introduce human bias.
3. Generality of Results: The experiments are limited to five LVLMs. It could be beneficial to see results on a broader range of models to generalize the findings better.

**Relation To Prior Work:**

The paper clearly discusses how this work differs from previous contributions and builds upon them effectively.

**Summary And Contributions:**

This paper introduces the VLKEB benchmark, which is designed to evaluate the knowledge editing methods for Large Vision-Language Models (LVLMs). The authors argue that existing benchmarks for LVLMs are limited by the quality of synthesized evaluation images and the lack of portability assessment. The proposed VLKEB addresses these issues by using real images and extending the portability metric. The benchmark also leverages a multi-modal knowledge graph to provide a more comprehensive evaluation framework. The paper includes experiments on various LVLMs, highlighting the strengths and weaknesses of different editing methods.

---

> ### Author Rebuttal · Authors · 2024-08-15
>
> Thank you for your feedback and evaluation. We noticed that the review titled "Its quality is **marginally above acceptance threshold**." was accompanied by a score of "5: Marginally below acceptance threshold." We believe this discrepancy may be due to differences in scoring criteria between the main track and the Datasets and Benchmarks track. We kindly request you to consider this potential misunderstanding when finalizing your assessment. Thank you for your consideration.
>
> **1. Data Source Limitations:**
>
> We appreciate the opportunity to respond to these concerns.
>
> **(i) Diversity and Coverage of Data As demonstrated** in Figure 5 of the Appendix, our dataset encompasses more than 20 entity categories (e.g., film, people, location, tv, organization, music, sports, education, government, business...). Furthermore, our benchmark has 8174 edit cases, 4819 portability tests and 18436 images, ensuring that our benchmark is sufficiently robust for the analyses we conduct.
>
> **(ii) Reason for Choosing MMKG** We selected MMKG due to its ability to provide correspondences between images and entities. This feature is crucial as it allows us to test portability metric. And our research is primarily focused on knowledge editing task, and MMKG serves as an ideal foundational data source for this purpose. Its structure and content are well-suited to our task designs, making it an appropriate choice for our studies.
>
> We believe these points highlight the appropriateness and effectiveness of choosing MMKG for our research goals.
>
> **2.	Manual Image Selection:**
>
> We appreciate the opportunity to clarify the procedures and rationale behind the manual image selection approach.
>
> **(i) We consider the efforts of manual selection part of our contribution, which can benefit subsequent research.** The manual selection process is crucial for ensuring that each image accurately corresponds with its designated entity, which is essential for accurate evaluation. This process not only guarantees the relevance and accuracy of our benchmarks but also contributes to the high quality of the dataset.
>
> **(ii) Initial Filtering** Before manual intervention, we employ a combination of filtering rules and the CLIP model to pre-filter the images. This preliminary step significantly reduces the volume of images, allowing human annotator to focus only on necessary verifications, thereby streamlining the process.
>
> **(iii) Bias Minimization Selection** During manual selection, annotators assess whether the images are similar yet distinct and whether the content correlates with the paired entities. Annotators do not judge the entities represented by the images nor exclude any based on bias. This focus on relevance and correspondence rather than subjective interpretation helps minimize the introduction of bias. We believe these steps ensure a rigorous approach to creating a useful and relatively less biased benchmark dataset.
>
> **3.	Generality of Results:**
>
> Thank you for your feedback on the generality of our results. We recognize the importance of testing across a broader array of models to enhance the generalizability of our findings. And we would like to explain why we chose these models.
>
> **(i) Model Selection Rationale** We selected five models that encompass diverse architectural designs to maximize the coverage of different approaches: BLIP employs a q-former (learned query) design, MiniGPT-4 incorporates both q-former and linear layers, LLaVA utilizes a linear projection approach, Qwen-VL features a learnable query combined with cross-attention, and mPLUG-Owl uses a visual abstractor (learnable query) and a modality adaptive module.
> These models were chosen to ensure a broad representation of existing designs, allowing us to draw useful conclusions from their differences. Additionally, the selected models vary in capability and size, which adds to the credibility of the results.
>
> **(ii) Future Expansion** While expanding our experiments to include more models would indeed be beneficial, our current resources and time constraints limit this capability. We plan to extend our analysis to additional models in future work, which should provide further insights and enhance the generalizability of our findings.
>
> We hope this clarifies the scope of our current study and our plans for expanding this research.
>
> **4.	Limitations:**
>
> Thank you for raising important concerns regarding the ethical considerations and potential societal impacts. We acknowledge that the open methods employed in our research could potentially be misused to distort the knowledge within multimodal large models, leading to toxic outputs. However, regarding the data, it is important to clarify that our research utilizes publicly available datasets that adhere to usage licenses, mitigating direct ethical issues related to the data itself. We will enhance our manuscript by incorporating discussion on these points.
>
> **5.	Future work and potential application of VLKEB benchmark.**
>
> We discussed future research directions in the last section based on the observations from experiments on our benchmark, e.g., finding direct LVLM editing methods and paying more attention to sequential editing and portability metrics. We expect VLKEB to be an effective benchmark that tests knowledge editing methods. And regarding the task of multimodal knowledge editing, it might be used in continual learning of LVLMs, where it can contribute to updating and refining the models' capabilities over time. This ongoing adaptation is crucial for maintaining relevance and accuracy as new data becomes available or as user needs evolve.

---

> ### Author Response · Authors · 2024-08-26
> **Friendly Reminder: Rebuttal Response for Discussion**
>
> Dear Reviewer Y4B2,
>
> We hope this message finds you well. As the deadline for the author-reviewer discussion approaching, we wanted to kindly follow up regarding our recent rebuttal. We understand that you may have a busy schedule, but we would greatly appreciate your feedback and any additional comments you might have.
>
> Following your valuable feedback, we have been working on integrating the LLaVA-OneVision model into our experiments. We have pushed a branch ([https://github.com/VLKEB/VLKEB/tree/dev-llava-ov](https://github.com/VLKEB/VLKEB/tree/dev-llava-ov)) with the initial integration which is under development to our GitHub repository, and we are continuing to refine and expand this work. We greatly appreciate your insights and are committed to addressing your concerns.
>
> Your insights are invaluable to us, and we are eager to continue the discussion to ensure our submission is as strong as possible. Please let us know if you need any further clarification or if there's anything we can do on our end.
>
> And again we kindly request you to consider the potential misunderstanding under the discrepancy between your review titled "marginally above acceptance threshold" and the score "5: marginally below acceptance threshold" when finalizing your assessment.
>
> Thank you for your time and consideration.

---

> ### Author Response · Authors · 2024-08-29
> **Awaiting Feedback on Rebuttal**
>
> Dear Reviewer Y4B2,
>
> Hope you are doing well. As the deadline for the author-reviewer discussion is very close, we wanted to gently remind you about providing your feedback on our rebuttal submitted on OpenReview. We greatly appreciate your insights and are looking forward to your comments.
>
> Thank you very much for your attention.
>
> Best regards,
>
> NeurIPS 2024 Datasets and Benchmarks Track Submission1032 Authors

---

### Author Rebuttal · Authors · 2024-08-17

### **General Response**

We would like to extend our deepest gratitude for the time, effort, and expertise that all the reviewers have dedicated to reviewing our manuscript. Your insights and suggestions are valuable to us and have provided clear directions for enhancing the quality of our research. Each point of feedback has been carefully considered, and where applicable, revisions have been made to address the concerns.

**Specific Revisions**

Following the suggestions, we have clarified the definitions and introductions of mathematical symbols to improve readability and understanding. Adjustments have been made to the figures to enhance their clarity and informativeness. Additional discussions on the ethical implications and societal impacts of our research have been incorporated to provide a comprehensive view of our work.

**Commitment to Continuous Improvement**

We plan to extend our experiments to additional LVLMs in future work. This expansion will not only enhance the generalizability of our findings but also enrich our understanding of the knowledge editing methods’ behavior across various architectures and settings. We are committed to exploring these opportunities in subsequent studies, guided by your insightful recommendations.

&nbsp;

We hope that our revisions and responses adequately address your concerns, and we look forward to any further suggestions you might have that could help improve our manuscript. We are enthusiastic about the potential of our work and its contribution to the field, and we thank you again for your crucial role in this process.

---

### Decision · Program_Chairs · 2024-09-26

**Decision:**

Accept (Poster)

**Comment:**

Most reviewers agreed that the proposed benchmark provides significant impact on the LVLM knowledge editing research, and the manuscript is well-structured. There are a few concerns raised, e.g., data source limitation and manual image selection, but the authors provided reasonable answers to these. We encourage the authors to reflect the rebuttals to their final paper.